# Global burden of Alzheimer's disease and other dementias attributable to smoking in 204 countries and territories, 1990–2021

Qifeng Tong[1,2], Jianing Wu[3], Qingchuan Jiao[1], Hao Wu[1], Jianqiu Gong[1], Qiang Wu[1]*

1 Department of Rehabilitation Medicine, Shaoxing People's Hospital, Shaoxing, Zhejiang, China, 2 School of Medicine, Shaoxing University, Shaoxing, Zhejiang, China, 3 Pelvic Floor Dysfunction Diagnosis, Treatment and Rehabilitation Center, Sir Run Run Shaw Hospital, Zhejiang University School of Medicine, Hangzhou, Zhejiang, China

* dr5rehab@163.com

## Abstract

### Background

Alzheimer's disease and other dementias (ADD) are significant global public health challenges. Smoking is a clearly established modifiable risk factor for dementia.

### Objective

This study aims to systematically elucidate the burden of ADD attributable to smoking from 1990 to 2021.

### Methods

We obtained data on Disability Adjusted Life Years (DALYs) and age-standardized DALYs rate (ASDRs) associated with ADD attributable to smoking from the Global Burden of Disease (GBD) database for the years 1990–2021. These data were disaggregated by gender, age, sociodemographic index (SDI), and region. Temporal trends in the burden of smoking-induced ADD were examined by calculating the average annual percentage changes.

### Results

From 1990−2021, global ASDR of smoking-attributed ADD declined 21.3% (EAPC = −0.88%) while DALYs increased 93% to 1.53 million. Females showed faster ASDR decline (EAPC = −1.50% vs male −0.73%). DALYs peaked at 65−85 years with accelerated crude rates post-75. Regionally, East Asia (611760.52, 262060.15−1401979.1), Western Europe (184593.67, 78898.94−420812.86), High-income North America (164283.74, 72809.79−373190.34) had highest 2021 DALYs; East Asia (29.95, 12.67−68.10), High income North America (23.32, 10.39−52.46), Tropical

**Data availability statement:** All date used in our research are available from the GBD2021 database (https://www.healthdata.org/research-analysis/gbd).

**Funding:** The author(s) received no specific funding for this work.

**Competing interests:** The authors have declared that no competing interests exist.

Latin America (20.81, 9.05–48.62) had highest ASDR. Nationally, China, United States of America and Lebanon led burdens. Steepest declines occurred in Mexico (EAPC = −3.14%), South Africa (−3.14%), and Sri Lanka (−2.84%). ASDR correlated with SDI (r = 0.44, p < 0.001) showing bimodal peaks at SDI = 0.5 and 0.8. Frontier analysis revealed peak heterogeneity in Alzheimer's disease ASDR around SDI 0.75, with the largest effectiveness disparities observed primarily in Middle-High SDI countries, indicating urgent needs for targeted health interventions despite declining trends in high-SDI nations.

## Conclusion

Despite declining dementia ASDR, global DALYs rose absolutely. The burden disproportionately impacted older populations, males, and high-middle-income nations. Mitigation requires context-adapted interventions including enhanced tobacco control, equitable healthcare access, and targeted health education.

## Introduction

According to the World Health Organization (WHO) 2021 forecasts, Alzheimer's disease and other dementias (ADD) are the seventh largest cause of mortality globally and a substantial contributor to disability among those aged 65 years and older worldwide [1]. It is a condition characterized by a spectrum of symptoms, including deficits in memory, language, problem-solving, and other mental capacities [2]. Alzheimer's disease (AD) is the most prevalent kind of dementia [3]. Total expenses for health care, long-term care, and hospice services for those aged 65 and older with dementia are estimated to reach $360 billion in 2024, with costs expected to top $1 trillion by 2050 [4,5]. The poor health and high expenses of care caused by dementia impose significant stress on families, communities, healthcare systems and social safety nets.

Several potentially modifiable risk factors, including smoking, contribute to dementia, as confirmed by observational studies [6]. Specifically, smoking raises the relative risk of dementia by as much as 1.6 times in later life (aged ≥65 years) and is the third highest contributing factor [7]. The detrimental effects of smoking on brain health are multifaceted. Key mechanisms linking smoking to ADD include: (i) Promotion of oxidative stress and neuroinflammation, damaging neurons and glial cells [8]; (ii) Induction of vascular damage and endothelial dysfunction, compromising cerebral blood flow and the blood-brain barrier integrity [9,10]; and (iii) Potential acceleration of amyloid-β (Aβ) peptide deposition and tau protein hyperphosphorylation, the hallmark pathological features of AD [11]. However, to date, there is no effective therapy for clinical dementia. Accumulating research has indicated that more than one-third of dementia cases might be avoided or postponed by the treatment of modifiable risk factors, including smoking [7]. Hence, research into dementia prevention is of critical relevance if the dementia pandemic is to be prevented.

Despite global progress in reducing smoking prevalence, critical gaps persist in implementing evidence-based tobacco control policies [12].

Critically, epidemiological studies reveal a significant upward trend in smoking-attributable AD mortality. This is evident in China (1990–2019), particularly among females [13], and is a global concern, exemplified by the Russian Federation's rapid increase versus Sri Lanka's decline among Belt and Road Initiative countries [14]. Moreover, this burden challenges countries across all income levels [15,16]. These disparities stem from systemic deficiencies: persistently low taxation sustaining affordability [17]; poor enforcement of advertising bans and smoke-free zones [18,19]; and chronic public health underfunding [20]. These systemic failures create a vicious cycle: low taxes enhance accessibility, regulatory lapses perpetuate exposure, and funding gaps hinder access to cessation services – collectively undermining dementia prevention and escalating smoking-related disease burdens. However, existing burden assessments face significant limitations. Most prior studies relied primarily on mortality metrics, failing to capture dementia's substantial non-fatal burden (e.g., DALYs imposing heavy socioeconomic and caregiver burdens) [13,16]. Furthermore, three key evidence gaps remain: (i) Reliance on outdated GBD data and mortality-centric metrics, inadequate for dementia; (ii) Insufficient granularity across socioeconomic, geographic, gender, and age dimensions; and (iii) The need to assess post-pandemic burden shifts using the latest GBD 2021 data [4,21].

Consequently, the global burden of AD attributable to smoking remains inadequately characterized, hindering targeted policy responses. Therefore, using GBD 2021 data, this study assesses the global burden and trends of smoking-related AD, with emphasis on variations across socioeconomic, geographic, gender, and age levels.

Therefore, leveraging the most recent and comprehensive GBD 2021 data, this study aims to provide a systematic assessment of the global burden and trends of smoking-related dementias. We specifically focus on DALYs and analyze their variations across socioeconomic development levels (via SDI), geographic regions, countries, sex, and age groups. This analysis is crucial for informing evidence-based tobacco control policies and dementia prevention initiatives tailored to diverse populations and settings.

## Methods

### Data source and collection

The University of Washington's Institute for Health Metrics Evaluation conducted the Global Burden of Disease (GBD) study, providing systematic assessments of 371 diseases and injuries in 204 countries/territories during 1990–2021. Estimates are stratified by age (0–95+), sex, location, and year. The GBD 2021 database includes incidence, prevalence, mortality, years of life lost (YLLs), years lived with disability (YLDs), and DALYs [22,23]. To ensure consistent burden estimates, the study employed the Disease Modeling–Meta Regression tool (DisMod-MR 2.1). This Bayesian-based framework synthesizes epidemiological data while accounting for established disease relationships and spatial patterns. Data for this analysis were retrieved from the Global Health Data Exchange (GHDx) query tool(https://vizhub.healthdata.org/gbd-results/). We extracted annual measurements of smoking-attributable ADD disability-adjusted life years (DALYs) and age-standardized DALY rates (ASDR) for the period 1990–2021. The data were stratified by age (0–95+ years), sex (female, male, both), and location (global, 5 SDI regions, 21 regions, and 204 countries/territories). The ASDR (per 100,000 population) was calculated using the GBD standard population structure to account for age composition differences and enable cross-year and cross-country comparisons. Additionally, we selected 12 age groups to examine DALY trends across various age groups, including a category for those aged ≥95 years and 5-year age groups spanning from 40 to 94 years.

All aggregated, de-identified datasets are publicly available in the GBD database, and no ethical approval was required for this secondary analysis.

### Measures of burden and data presentation

DALYs and the DALYs rate from 1990 to 2021 are two indicators of the disease burden for ADD. A summary metric for calculating the total burden of disease is the DALY.

DALYs quantifies the total health loss attributable to a disease or risk factor. One DALY equals one lost year of healthy life and is the sum of two components: Years of life lost due to premature mortality (YLL) and Years lived with disability (YLD). The Definitions: YLL = (standard life expectancy at age of death) × (number of deaths); YLD = (number of prevalent cases) × (disability weight) × (average duration of disability). We provide a simple example to help you understand: Consider a 50-year-old who develops AD because of smoking and dies at age 65. If the national reference life expectancy is 80 years:

- YLL = 80 − 65 = 15 years

- YLD: from age 50–65 the patient lives 15 years with AD. Using the GBD 2021 disability weight for moderate dementia (0.449): YLD = 15 × 0.449 = 6.7 years

- Total DALYs for this individual = 15 + 6.7 = 21.7 years

Thus, DALYs capture both the mortality gap (YLL) and the burden of living with disability (YLD). Real-world estimates are derived from complex models, but the principle remains the same: DALYs convert adverse health events into a single metric of lost healthy time. The modeling approach used by GBD 2021 to estimate DALY has been thoroughly explained elsewhere [22,24].

Countries were stratified into five Socio-demographic Index (SDI) quintiles (low to high) based on a composite metric integrating three development dimensions: (i) education (mean years schooling ≥15 years); (ii) fertility (total rate <25 years); (iii) income (lag-distributed per capita) [25]. In addition, the globe is further divided into 21 areas according to physical closeness and epidemiological similarities.

## Definitions

According to GBD 2021, dementia is a chronic, progressive, degenerative neurological condition marked by cognitive dysfunctions that make it difficult to perform daily tasks.

We use the Diagnostic and Statistical Manual of Mental Disorders III, IV, or V, or ICD case definitions as the reference. The condition is classified as a distinct clinical entity under the International Classification of Diseases, 11th Revision (ICD-11; code 6D80) [26].

Smoking was defined as the prevalence of current use of any smoking tobacco product and the prevalence of previous use of any smoking tobacco product. It shows the number of cigarettes smoked daily and the total number of years of exposure for smokers who are currently smoking, and it estimates the distribution of years since quitting for smokers who have previously smoked [27].

## Statistical analysis

We analyzed ADD trends and burden between 1990 and 2021 using a variety of statistical methods. First, we described the number and ASDR of ADD attributable to smoking. The direct standardization method was used to calculate the age-standardized rate (ASR), and the weights were applied according to the GBD 2021 world standard population. ASDRs were reported per 100,000 and stratified by region, sex, year, and age group. The data were presented as numerical values, along with their 95% CIs or 95% UIs.

To visualize the geographic variation in dementia ASDR due to smoking and its estimated annual percentage change (EAPC) in 2021, we generated a map of ADD due to smoking. In order to assess the association between smoking-induced ADD burden and SDI and ascertain the influence of sociodemographic factors on this burden, the Pearson correlation test, Gaussian process regression model in conjunction with the Loess smoother, and Spearman rank order correlation test were utilized.

The average increased or decreased rate for the change of a certain variable during a given period is represented by average annual percentage changes, or AAPCs. According to the underlying joinpoint regression model, the weighted

average of the slope coefficient was converted into the yearly change percentage for this study from 1990 to 2021 [28]. Joinpoint regression was used to estimate AAPCs from the aforementioned ASDR with 95% CI in order to quantify the temporal trend. During 1990–2021, the value of AAPCs showed a rise (+), fall (−), or no change (0) in the percent yearly change. The comparable rate was trending higher if AAPCs were greater than 0. The comparable rate was trending downward if AAPCs were less than zero. The equivalent rate remained constant otherwise.

The least achievable illness burden for each nation or area, given its existing SDI level, may be found using frontier analysis. By determining the lowest load that might be theoretically attained given the existing SDI, this method acts as a benchmark for optimal frontier performance. This method identifies leading countries and territories, setting standards and targets for others. The "effective differences," which show the difference between the current illness burden and the lowest potential burden, were calculated for each location. Specifically, effectiveness differences are equal to the value of the ASDR for each country minus the value of the frontier, adjusted for SDI. These variations offer a numerical indicator of each nation's or region's distance from the ideal performance standard.

R software (R Core Team, version 4.3.2, Vienna, Austria) and the Joinpoint Regression Program (version 5.2.0) were used for all of the aforementioned studies.

## Results

### Global burden of ADD attributable to smoking from 1990 to 2021

The worldwide burden of ADD changed significantly between 1990 and 2021. A thorough comparison of DALYs and ASDR by SDI levels and GBD areas throughout this time period, including EAPC data, is shown in Table 1.

Globally, the number of ADD DALYs increased from 794.92 thousand to 1533.21 thousand, and the ASDR declined by 21.3%, from 23.33 to 18.36 per 100,000. An overall downward trend was shown by the ASDR's EAPC of −0.88% (Table 1).

With the increase of age, the number of DALYs of males and females in the global first increased and then decreased, and the peak was 65–85 years old. The global crude DALYs rate for males and females continued to rise, rising slowly before the age of 75 and accelerating after the age of 75. The number of DALYs in females is greater than that in males, while the crude DALYs rate in males is higher than that in females (Fig 1, S1 and S2 Tables).

In 21 global regions, Records were recorded in East Asia (611760.52, 262060.15–1401979.1), Western Europe (184593.67, 78898.94–420812.86), and High-income North America (164283.74, 72809.79–373190.34) had the highest number of DALYs associated with ADD caused by smoking in 2021. In contrast, the lowest numbers were reported in Oceania (661.09, 279.48,1524.18), Central Sub-Saharan Africa (2170.14, 823.42,5189.26), and Andean Latin America (3187.59, 1352.82,7589.45). For ASDR, East Asia (29.95, 12.67–68.10), High income North America (23.32, 10.39–52.46), Tropical Latin America (20.81, 9.05–48.62) has a higher rate, Western Sub-Saharan Africa (2.95, 1.15–7.14), Central Sub-Saharan Africa (5.05, 1.93–12.35) and Andean Latin America (5.65, 2.40–13.40) are lowest.

Among countries with different levels of SDI, low SDI countries had the lowest number of DALYs (30071.64, 11977.78–72619.96) and ASDR (8.1, 3.29–19.44) related to smoking. The number of DALYs in Middle SDI countries was the highest (470384.96, 201779.14–1086306.44), while the ASDR value in High-middle SDI countries was the highest (21.96, 9.6–49.15).

From 1990 to 2021, the ASDR values in most regions showed a downward trend, Southern Saharan Africa (EAPC = −2.65%, −2.74 to −2.56), Central Latin America (EAPC = −2.25%, −2.36 to −2.14) and Tropical Latin America (EAPC = −1.87%, −2.05 to −1.69) showed the most significant decrease (Table 1).

In 2021, among the 204 countries globally, China (602501.08, 257945.38 to 1379582.96), United States of America (148705.22, 65330.84 to 336695.05), and India (98343.86, 40167.81 to 236215.30) had the highest number of DALYs attributable to smoking-related ADD. The countries with the highest ASDR were Lebanon (35.63, 15.39 to 80.46), China (30.63, 12.98 to 69.65), and Albania (30.54, 12.98 to 69.92) (Fig 2A, S3 Table).

**Table 1. DALYs and ASDR for Alzheimer's disease and other dementias attribute to smoking for both sexes, 1990–2021.**

| Location | 1990 | | 2021 | | 1990-2021 |
|---|---|---|---|---|---|
| | DALYs | ASDR (per 100,000) | DALYs | ASDR (per 100,000) | EAPC of ASDR |
| | (95% UI) | /10⁵ (95% UI) | (95% UI) | /10⁵ (95% UI) | (95% UI) |
| **Global** | 794915.2 (344377.51,1839709.42) | 23.33 (9.99,54.46) | 1533213.54 (662722.71,3496419.97) | 18.36 (7.9,42.07) | −0.88 (−0.92, −0.83) |
| **Female** | 262309.62 (112857.3,602150.56) | 13.65 (5.8,31.32) | 423190.66 (185084.81,947536.47) | 9.01 (3.94,20.16) | −1.5 (−1.56, −1.45) |
| **Male** | 532605.58 (227758.66,1241825.38) | 37.45 (15.98,87.69) | 1110022.88 (474038.47,2558503.5) | 30.56 (12.72,71.5) | −0.73 (−0.8, −0.66) |
| Low SDI | 15214.67 (6575.47,36753.83) | 9.72 (4.09,23.07) | 30071.64 (11977.78,72619.96) | 8.1 (3.29,19.44) | −0.64 (−0.69, −0.59) |
| Low-middle SDI | 71967.31 (31552.49,171280.44) | 15.95 (6.82,37.64) | 153656.95 (63420.52,366741.77) | 13.02 (5.45,31.35) | −0.7 (−0.72, −0.68) |
| Middle SDI | 195492.62 (83542.77,446430.43) | 24.14 (10.01,56.67) | 470384.96 (201779.14,1086306.44) | 19.38 (8.25,44.76) | −0.88 (−0.94, −0.82) |
| High-middle SDI | 200798.18 (84592.04,461915.08) | 22.72 (9.46,52.39) | 435095.67 (191159.58,971668.74) | 21.96 (9.6,49.15) | −0.17 (−0.19, −0.15) |
| High SDI | 310666.04 (136518.8,716664.77) | 27.97 (12.27,64.53) | 442915.27 (190964.64,1020743.64) | 19.05 (8.33,43.29) | −1.34 (−1.4, −1.28) |
| **Central Europe, Eastern Europe, and Central Asia** | 59977.01 (26155.13,136934.54) | 13.36 (5.88,30.28) | 85194.66 (36806.43,194785.15) | 12.77 (5.54,29) | −0.06 (−0.23,0.12) |
| Central Europe | 26911.11 (11851.07,61193.25) | 19.07 (8.36,43.19) | 33947.2 (14976.68,77856.86) | 14.77 (6.44,33.75) | −0.84 (−0.9, −0.78) |
| Eastern Europe | 28706.08 (12568.17,65937.25) | 10.81 (4.76,24.69) | 43177.68 (18713.43,97784.86) | 11.92 (5.23,26.87) | 0.43 (0.16,0.7) |
| Central Asia | 4359.82 (1950.11,10094.91) | 10.16 (4.53,23.61) | 8069.78 (3529.27,18936.2) | 11.45 (5.08,26.38) | 0.74 (0.6,0.87) |
| **High-income** | 794915.2 (344377.51,1839709.42) | 23.33 (9.99,54.46) | 1533213.54 (662722.71,3496419.97) | 18.36 (7.9,42.07) | −1.41 (−1.46, −1.36) |
| Australasia | 4412.13 (1974.59,10040.35) | 18.86 (8.37,42.69) | 7015.67 (3016.71,16579.78) | 12.11 (5.19,28.39) | −1.4 (−1.46, −1.34) |
| Asia Pacific | 45858.15 (19474.86,106212.34) | 24.84 (10.44,57.2) | 89722.29 (38568.94,210580.36) | 15.81 (6.77,36.48) | −1.57 (−1.62, −1.51) |
| North America | 125526.66 (55483.87,291114.14) | 34.48 (15.23,79.44) | 164283.74 (72809.79,373190.34) | 23.32 (10.39,52.46) | −1.4 (−1.47, −1.33) |
| Western Europe | 154412.36 (67714.32,351220.85) | 25.78 (11.25,59.3) | 184593.67 (78898.94,420812.86) | 17.23 (7.43,38.39) | −1.34 (−1.38, −1.31) |
| Southern Latin America | 5659.5 (2542.55,12844.68) | 12.59 (5.64,28.39) | 9343.13 (4054.71,20860.4) | 10.53 (4.59,23.53) | −0.56 (−0.66, −0.46) |
| **Latin America and Caribbean** | 335868.8 (147812.41,775053.15) | 27.67 (12.13,63.87) | 452670.05 (193416.97,1043520.29) | 18.37 (7.98,41.6) | −1.82 (−1.93, −1.71) |
| Andean Latin America | 1261.62 (530.2,2998.35) | 7.05 (2.96,16.84) | 3187.59 (1352.82,7589.45) | 5.65 (2.4,13.4) | −0.84 (−0.93, −0.74) |
| Caribbean | 3466.03 (1490.12,7883.97) | 14.49 (6.29,33.31) | 6059.5 (2666.38,14005.37) | 11.08 (4.89,25.64) | −1.06 (−1.19, −0.93) |
| Tropical Latin America | 26317.5 (10754.81,62011.3) | 34.9 (14.37,82.59) | 52048.25 (22656.24,122054.32) | 20.81 (9.05,48.62) | −1.87 (−2.05, −1.69) |
| Central Latin America | 8966.49 (3991.71,20343.85) | 12.89 (5.72,29.41) | 9343.13 (4054.71,20860.4) | 6.74 (2.97,14.9) | −2.25 (−2.36, −2.14) |
| **North Africa and Middle East** | 30637.27 (13260.93,69329.98) | 23.87 (10.23,54.52) | 66396.79 (28430.2,151668.29) | 18.22 (7.72,41.54) | −0.96 (−1.01, −0.9) |

*(Continued)*

**Table 1.** (Continued)

| Location | 1990 | | 2021 | | 1990-2021 |
| --- | --- | --- | --- | --- | --- |
| | DALYs | ASDR (per 100,000) | DALYs | ASDR (per 100,000) | EAPC of ASDR |
| | (95% UI) | /10⁵ (95% UI) | (95% UI) | /10⁵ (95% UI) | (95% UI) |
| **South Asia** | 55863.97 (24134.04,134194.63) | 13.94 (5.84,32.94) | 129918.09 (52973.46,310723.34) | 10.95 (4.55,26.31) | −0.84 (−0.88, −0.81) |
| **Southeast Asia, East Asia, and Oceania** | 260201.98 (109095.35,597368.39) | 30.25 (12.14,70.95) | 702143.9 (303832.62,1609939.54) | 27.07 (11.47,61.94) | −0.48 (−0.54, −0.41) |
| Southeast Asia | 40981.23 (18278.54,93081.41) | 20.78 (9.1,47.13) | 89722.29 (38568.94,210580.36) | 16.17 (6.85,37.31) | −0.99 (−1.09, −0.89) |
| East Asia | 218922.24 (90301.58,508008.64) | 33.68 (13.33,78.42) | 611760.52 (262060.15,1401979.1) | 29.95 (12.67,68.1) | −0.48 (−0.58, −0.39) |
| Oceania | 298.51 (124.32,699.01) | 12.78 (5.38,29.87) | 661.09 (279.48,1524.18) | 10.53 (4.46,24.93) | −0.73 (−0.79, −0.68) |
| **Sub-Saharan Africa** | 12354.53 (5187.16,28583.75) | 8.1 (3.29,19.44) | 19407.89 (7674.74,46960.78) | 5.27 (2.1,12.71) | −1.49 (−1.59, −1.38) |
| Central Sub-Saharan Africa | 964.29 (390.86,2326.23) | 5.88 (2.41,14.34) | 2170.14 (823.42,5189.26) | 5.05 (1.92,12.35) | −0.39 (−0.66, −0.12) |
| Eastern Sub-Saharan Africa | 4798.05 (1952.78,11610.54) | 9.54 (3.88,22.72) | 8554 (3336.71,21035.32) | 6.91 (2.76,16.31) | −1.17 (−1.26, −1.07) |
| Southern Sub-Saharan Africa | 4086.68 (1794.13,9195.33) | 18.6 (8,41.45) | 4103.91 (1728.64,9565.99) | 8.43 (3.5,19.63) | −2.65 (−2.74, −2.56) |
| Western Sub-Saharan Africa | 2505.52 (1013.64,5965.13) | 3.72 (1.48,8.68) | 4579.83 (1793.79,11406.07) | 2.95 (1.15,7.14) | −0.74 (−0.76, −0.72) |

From 1990 to 2021, the ASDR attributable to smoking-related ADD decreased in the majority of countries (Fig 2B). Mexico (EAPC = −3.14%, −3.32 to −2.96), South Africa (EAPC = −3.14%, −3.27 to −3.01), and Sri Lanka (EAPC = −2.84%, −3.11 to −2.57) exhibited the most substantial declines in ASDR (S4 Table).

## Temporal joinpoint analysis

Stratifying by age, we observed an overall gradual decline in crude DALYs rate across all age groups worldwide as the years increase, with larger age groups showing a faster decline, with a faster decline before 2010. In addition, because crude DALYs rate were higher in the older age group, the fluctuation level of decline was also greater in younger age groups (Fig 3A–D, S5–S7 Tables).

Fig 4AC shows the change in the burden of ADD from 1990 to 2021 by different SDI levels and by sex. From 1990 to 2021, ASDR demonstrated an overall declining trend across all SDI countries and sexes, albeit with fluctuations. High-SDI countries exhibited the most rapid decline, while middle and high-middle SDI countries experienced declining rates until 2019, followed by a reversal to an upward trend – particularly pronounced among males.

Overall, from 1990 to 2021, ASDR from smoking use-induced ADD in low-middle SDI countries and low SDI countries showed a relatively slow decline. However, the ASDR of females in low SDI countries showed a slight fluctuation from 1990 to 2016, which first increased and then decreased (S8–S10 Tables).

## Relationships between the SDI and the disease burden of ADD

The relationship between the SDI and ASDRs of smoking related ADD exhibited a significant positive correlation, respectively (r = 0.439, $p < 0.001$; r = 0.419, $p < 0.001$).

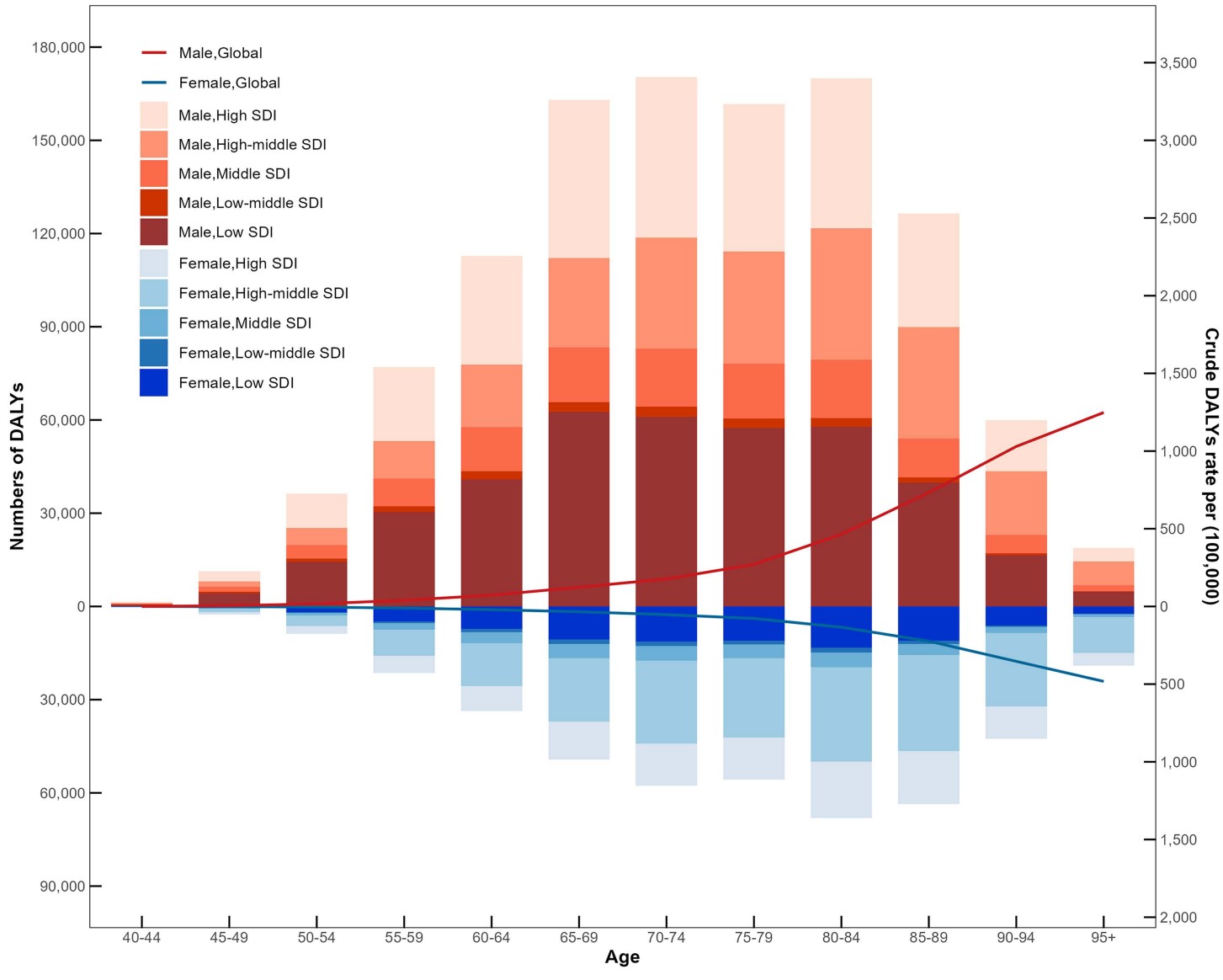

**Fig 1. Global age-standardized rate trends for smoking-related Alzheimer's disease and other dementias by age group in 1990 and 2021.**

Across all SDI regions from 1990 to 2021, the relationship showed a bimodal fluctuation upward trend; the ASDRs for ADD initially increased with rising SDI, peaking around an SDI of 0.5, before starting to decline. Subsequently, starting from SDI greater than 0.6, the ASDR value rises again, reaching a second peak at about 0.8, and finally decreasing (Fig 5A). This bimodal pattern, particularly the secondary peak in high-SDI regions, highlights the substantial heterogeneity in ASDR levels among countries with similar socioeconomic status. Importantly, this heterogeneity coexists with the overall declining temporal trends within SDI groups throughout the study period.

In 204 countries and territories, the burden of smoking-attributable ADD was typically positively connected with SDI at the national level in 2021. With the increase of SDI, ASDR rises gently at first. When SDI is about 0.8, ASDR rises rapidly (Fig 5B).

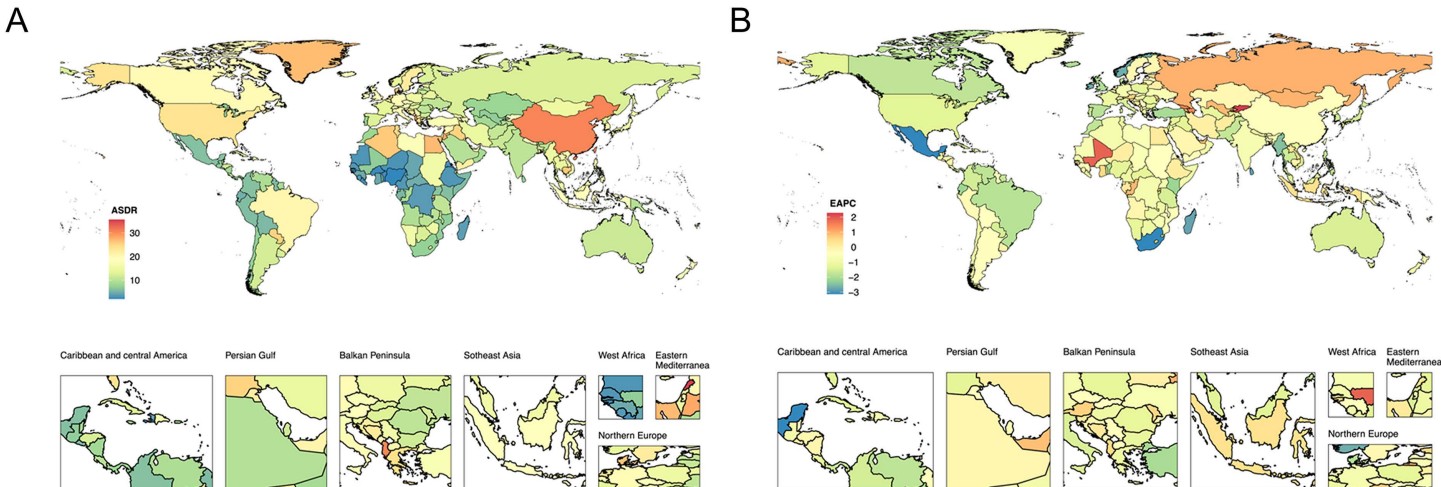

**Fig 2. The burdens of ADD associated with smoking among 204 countries and territories.** (A) ASDRs of ADD associated with smoking in 2021. (B) EAPC of ASDR for ADD Associated with Smoking (1990 - 2021).

## Frontier analysis involving the SDI and ADD burden

The ASDR for ADD in several nations from 1990 to 2021 is shown against SDI in Fig 5C. As a standard for comparison, the frontier line shows the expected ASDR based on SDI. The line-to-point difference (effective difference) represents the potential that countries could theoretically improve. As SDI increases, the instability of the frontier trend reflects that heterogeneity peaks around SDI 0.75, and subsequently declines in high-SDI regions. This pattern aligns with Fig 5A, which shows that after SDI 0.8, Western Europe, high-income Asia Pacific, and high-income North America maintain similar rapid downward trajectories in ASDR.

Fig 5D highlights disparities in ASDR among countries with similar SDI levels in 2021. The ASDR was low, and the gap was small among low SDI countries, such as Somalia (SDI = 0.08, ASDR = 5.80 per 100,000), Niger (SDI = 0.17, ASDR = 2.92 per 100,000), and Burkina Faso (SDI = 0.29, ASDR = 2.31 per 100,000). The top 15 countries with the largest differences in effectiveness were mostly with Middle-High SDI countries and a small number of High SDI countries, ranging from 24.04 to 35.63, indicating the urgent need for health interventions. Notably, middle SDI countries with the largest EF – Rwanda (SDI = 0.44, ASDR = 26.26 per 100,000), Cambodia (SDI = 0.47, ASDR = 26.06 per 100,000), Kiribati (SDI = 0.53, ASDR = 28.21 per 100,000), Egypt (SDI = 0.61, ASDR = 27.07 per 100,000), and Albania (SDI = 0.71, ASDR = 30.54 per 100,000) – had ASDRs increased, reflecting significant public health challenges. However, countries with high SDI, which have the largest EF – Greece (SDI = 0.79, ASDR = 24.25 per 100,000), Greenland (SDI = 0.83, ASDR = 26.93 per 100,000), United States of America (SDI = 0.86, ASDR = 23.75 per 100,000), Denmark (SDI = 0.90, ASDR = 26.39 per 100,000), Netherlands (SDI = 0.89, ASDR = 24.04 per 100,000), Sweden (SDI = 0.89, ASDR = 22.24 per 100,000), and Republic of Korea (SDI = 0.89, ASDR = 20.98 per 100,000) – showed a decline in ASDRs, suggesting relatively effective healthcare interventions (S11 and S12 Tables).

## Discussion

This study identified an overall gradual decline in crude DALYs rate across all age groups worldwide with increasing years. We also observed significant age-, sex-, and region-specific heterogeneity in the disease burden of ADD. Although the global ASDR showed an overall declining trend (EAPC = −0.88%), divergent patterns emerged across SDI regions: high-SDI regions achieved substantial burden reduction (EAPC = −1.34%), while middle SDI countries exhibited

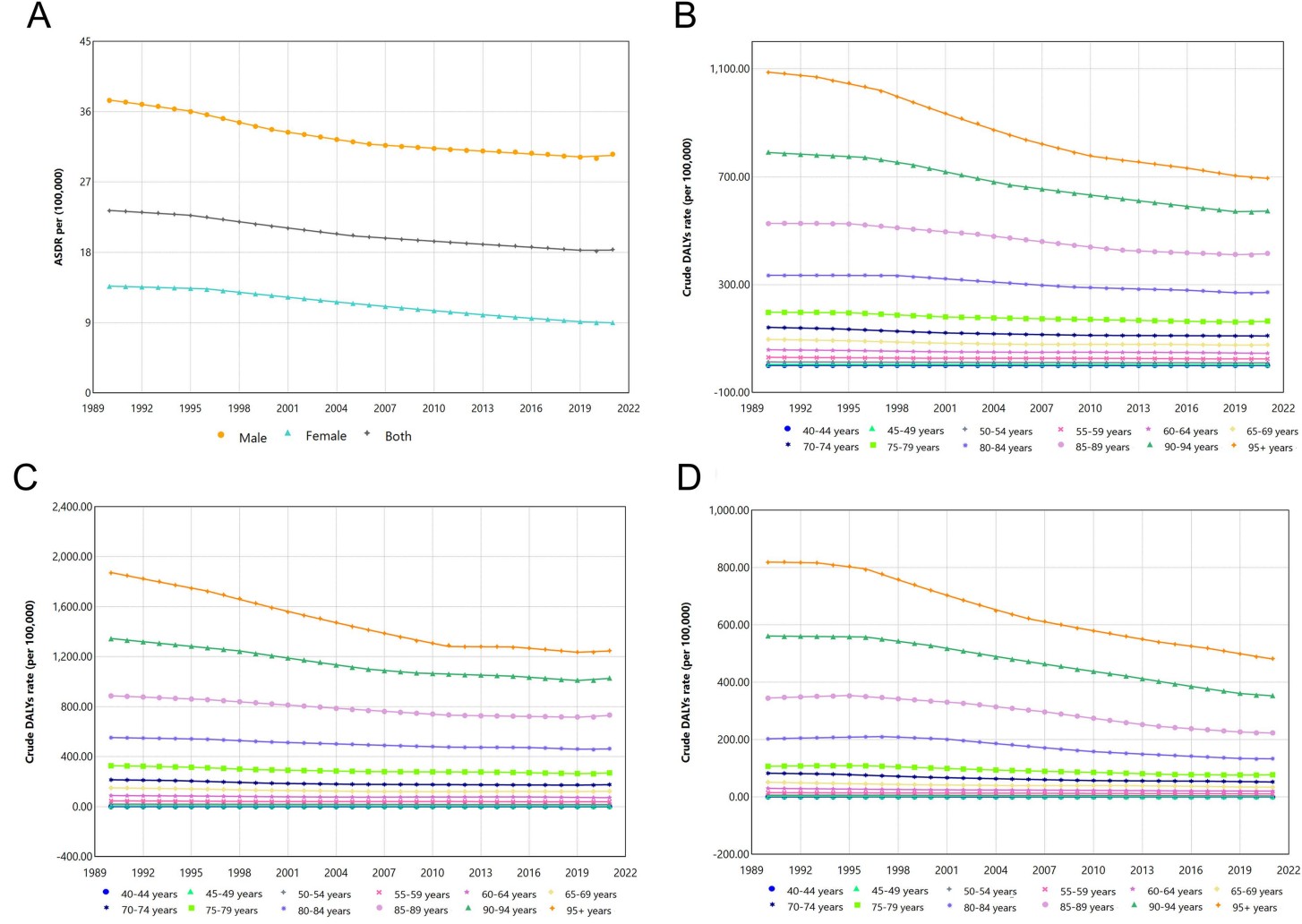

**Fig 3. Global Alzheimer's disease and other dementias DALYs rates attributable to smoking, 1990 - 2021.** (A) Global ASDRs by sex group; (B) Global both sexes by age group; (C) Global male by age group; (D) Global female by age group.

a modest decline in burdens (EAPC = −0.17%). The burden peaked in individuals aged 65–90 years, with systematically higher risks in males than females – a disparity most pronounced in low SDI settings. Notably, significant ASDR disparities persist even among middle to high SDI countries with comparable development levels, suggesting that healthcare policy implementation efficacy may exert greater practical influence than economic status alone in shaping dementia burden patterns.

### Temporal trends of smoking-related ADD

This is thanks to the WHO Framework Convention on Tobacco Control (FCTC) came into force in 2005 which providing parties with a comprehensive strategy to combat the tobacco epidemic and setting out a wide range of evidence-based measures to reduce the demand and supply of tobacco [29]. The influence on tobacco use may be assessed from the standpoint of tobacco control using the price elasticity of tobacco products. Tobacco price increases prompt consumers to reassess smoking's health and financial costs, driving nicotine reduction or cessation that subsequently lowers AD

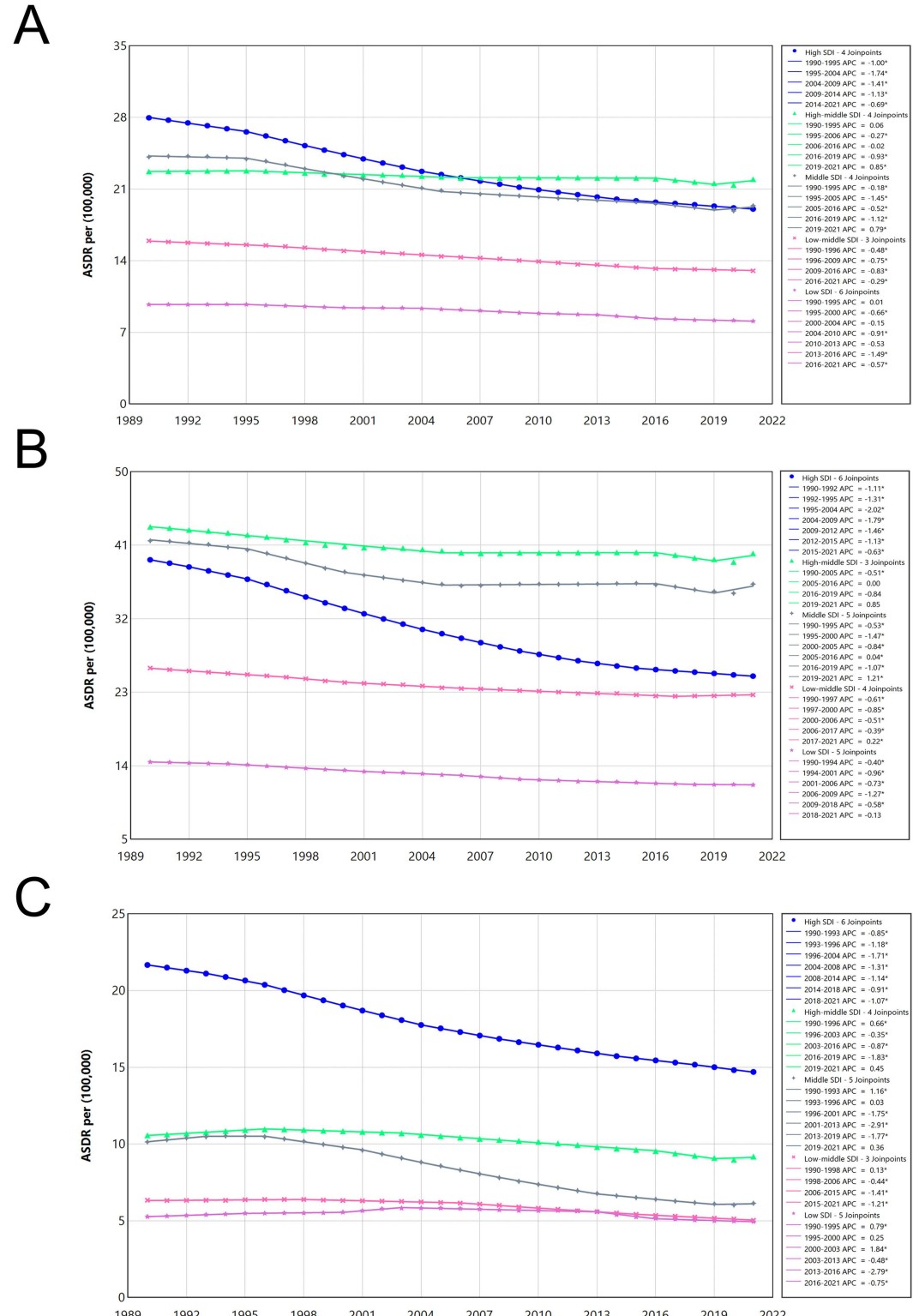

**Fig 4. Global Alzheimer's disease and other dementias ASDRs attributable to smoking by SDI regions group, 1990 - 2021.** (A) Global both sexes; (B) Global male; (C) Global female.

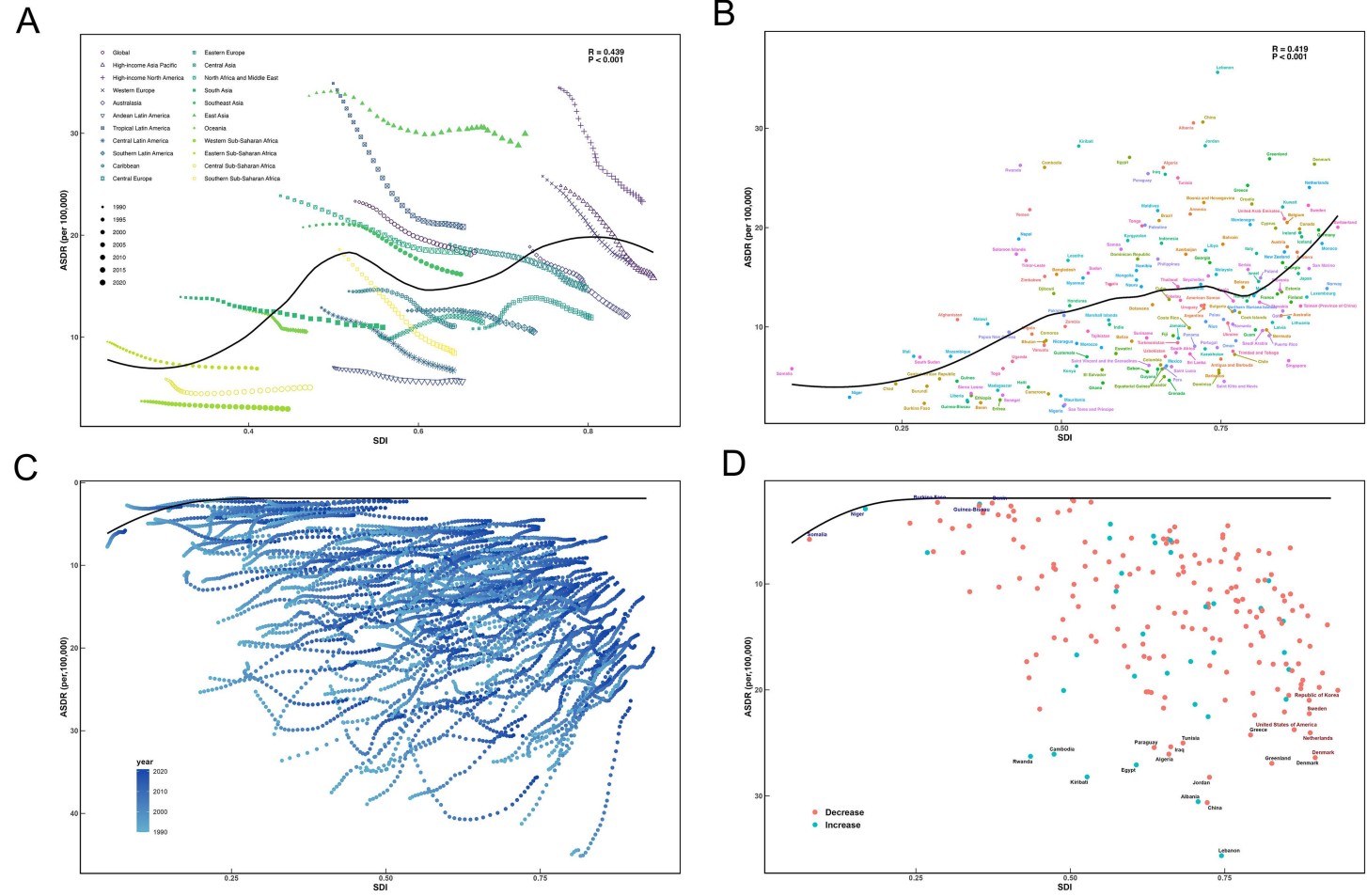

**Fig 5. The association between the Socio-Demographic Index and ASDRs of smoking-related ADD.** (A) The association between ASDRs of ADD related to smoking and SDI among 21 GBD regions. (B) The association between ASDRs of ADD related to smoking and SDI among 204 countries and territories. (C) Frontier analysis of global ASDR trends in ADD from 1990 to 2021. The color gradient indicates the year of the data, with darker colors representing more recent years. (D) Frontier analysis of ADD with ASDRs in 2021. The frontier is marked in solid black, and countries and territories are presented as dots. The 15 leading countries with the most effective differences (the highest ASDR of ADD gap from the frontier) are marked in black. Red dots represent a reduction in ADD burden between 1990 and 2021. Blue dots represent an increase in ADD burden during the same duration, and those countries are marked in green.

risk by mitigating key pathological pathways such as oxidative stress, neuroinflammation, and accelerated tau pathology [30]. Sustained progress requires persistent tobacco control implementation and optimized healthcare resource distribution.

The growing spatiotemporal heterogeneity in policy interventions is becoming increasingly evident. Mexico exemplifies this through its integrated "tobacco tax-cessation service" strategy, achieving accelerated ASDR reduction (SDI = −3.14%) [31]. Evidence confirms that tobacco taxation significantly reduces smoking-related disease burdens while generating fiscal space crucial for development financing in low and middle income countries [32]. While most countries have raised tobacco taxes under FCTC guidance, politically and economically fragile states often keep taxes low and cigarettes cheap. This disparity necessitates that the global tobacco control community prioritize targeted assistance to enhance FCTC compliance in vulnerable nations through a dual approach: building technical-administrative capacities while reinforcing institutional governance frameworks [33].

## SDI correlation and Frontier analysis

Our results indicate that overall ASDR rises with SDI, consistent with prior studies [16]. Additionally, a bimodal distribution pattern (peaking at SDI 0.5 and 0.8) was observed, potentially reflecting distinct risk mechanisms across development stages. Frontier analysis identified regions and nations with potentially suboptimal healthcare service efficiency, suggesting opportunities for targeted intervention and resource allocation.

Low middle SDI countries, such as Rwanda (SDI = 0.44, ASDR = 26.26/100,000) and Nepal (SDI = 0.43, ASDR = 18.86/100,000), may be associated with industrialization [34]. These could include inadequate tobacco tax, weak monitoring, partial implementation of advertising bans, and insufficient smoke-free areas and cessation programs weaken tobacco control policies – factors that compound the challenge of smoking-attributable AD in developing economies [15,35]. Such policy gaps, particularly when concurrent with developing health systems, might contribute to observed ASDR patterns [36,37]. Similarly noteworthy is Egypt's ASDR (SDI = 0.61, ASDR = 27.07/100,000), which appears elevated relative to comparable middle SDI countries. This pattern coincides temporally with a marked increase in male smoking prevalence, escalating from 34.2% in 2000 to a projected 62.9% by 2025 [38,39]. Of particular concern is the persistent stagnation of health investment in Egypt, with total health expenditure remaining static at 5.5% of GDP over the past 12 years – a figure substantially below WHO-recommended benchmarks for sustainable health financing [40]. Collectively, these observations show preliminary alignment with a conceptual model suggesting that rapid economic development coupled with insufficient health investment could potentially correlate with observed lags in health outcomes in certain contexts. This is particularly relevant for AD burden linked to smoking in resource-limited settings. Critically, investments should specifically include strengthening primary care capacity for early identification of cognitive decline in smokers and providing accessible cessation support –essential steps for mitigating this growing challenge in developing countries, thereby interrupting the neurotoxic cascade before significant pathology develops.

Another key barrier is the limited diagnostic capacity for AD in low and middle SDI countries: accurate diagnostic tools like Amyloid PET, FDG-PET, Tau PET, and CSF biomarkers are not easily accessible. AD screening techniques, treatments, and diagnoses are not widely available. Dementia remains misunderstood as normal aging, compounded by socio-economic disparities, culturally rooted stigma, and inadequate national policy frameworks in most countries [41].

ASDRs demonstrate a progressive elevation pattern across SDI, with high-SDI populations experiencing accelerated ADD burdens compared to disadvantaged countries. This trend was also reflected in ASMR [16]. Despite annual declines in smoking prevalence across high-SDI countries, the delayed neuropathological effects of tobacco use persistently contribute to elevated ASDR [42,43]. Additionally, demographic pressures from population aging and growth strain medical budgets in these regions. In certain high-SDI settings, extended survival of ADD patients may inadvertently elevate YLDs and contribute to a distinct secondary peak in ASDR observed in these regions [44]. Analysis of nations with dementia-friendly policies – such as Sweden (SDI = 0.89; ASDR = 22.24/100,000), the United Kingdom (SDI = 0.86; ASDR = 18.18/100,000), and Germany (SDI = 0.90; ASDR = 19.76/100,000) – suggests that this paradoxical burden pattern may arise from persistent implementation challenges: (i) Prolonged diagnostic intervals (21–70 months across settings) that exceed clinical targets, persisting even in contexts with adequate specialist density (e.g., Sweden's 13.6/100,0000) [45]; (ii) System disconnects between resource availability and care coordination, evidenced by regional fragmentation [45], specialist shortages [46], and insurance-based disparities [47]; and (iii) Suboptimal utilization of diagnostic technologies, particularly PET-dependent pathways [46] where biomarker access limitations may account for ~30% of delays [47]. These patterns suggest that diagnostic bottlenecks may enable disease progression before treatment eligibility, potentially explaining the elevated dementia burden observed in well-resourced systems. While biomarker-assisted diagnosis is guideline-recommended, adoption remains inconsistent [41,45–47]. Without a streamlined diagnostic pathway coordination – particularly to address delays in early detection, high-SDI countries risk accumulating preventable dementia burden despite substantial resources.

Bimodal curves of SDI and ASDRs reveal different contradictions in development stages: tobacco consumption was loosened in early industrialization, while advanced economies face health challenges from aging populations [48,49].

## Patterns of sex differences in disease burden

Smoking-related ADD poses a significant disease burden with striking gender disparities. It is generally believed that the prevalence and mortality of ADD are higher in females than in males [4,50,51]. However, our study found that number of DALYs and crude DALYs rate from smoking attributable ADD in global males were higher than in females. This divergence is primarily driven by sustained male-dominant smoking patterns [12,42], manifested behaviorally as men typically smoking more cigarettes per day, using higher-nicotine products, and inhaling more deeply than women [52]. These behaviors generate prolonged and intensified neurotoxic exposure (via oxidative stress, inflammation, and direct Aβ/tau pathology. Nevertheless, females face compensatory dementia risks through substantial secondhand smoke exposure [53,54], further amplified by the sharp post-menopausal decline in neuroprotective estrogen, which promotes neurogenesis and enhances Aβ clearance [55]. Moreover, protective factors operate asymmetrically—physical activity confers stronger cognitive resilience in females than males [56–58], whereas dietary patterns (particularly low B12/high-fat diets) selectively elevate AD risk in males through vascular-inflammatory pathways [59].

Although males had higher baseline ASDRs (Female 13.65/100,000 vs. Male 37.45/100,000) in 1990, our findings are consistent with previous studies [14] showing that the annual rate of change in ASDRs is declining more sharply in females than males worldwide (Female −1.5% vs. Male – 0.73%) (Table 1). This may be attributed to the greater decline in smoking prevalence among females (−37.9%) than males (−27.2%) under tobacco control policies. Previous research has demonstrated that female dementia patients are more likely to decide to quit smoking after learning about the health risks associated with smoking, while male dementia patients are more likely to continue smoking than their female counterparts due to tobacco dependence and increased social and environmental pressures [60]. Generally, these mechanisms underscore the urgency for sex-tailored interventions targeting specific pathways—prioritizing male-centric cessation programs (with antioxidant/vascular support/ dietary interventions like B12 and folate fortification), vascular risk mitigation, while advancing female-focused secondhand smoke legislation and hormone-modulating strategies.

Pathologically, chronic cigarette smoke induces sustained cerebral oxidative stress (OxS) through reactive oxygen species (ROS) overload and hypoxia, which directly initiates Alzheimer's pathology by activating β-secretase (promoting Aβ aggregation) and inducing tau hyperphosphorylation [61]. This OxS further triggers self-amplifying neuroinflammation via microglial dysfunction and cytokine release, which impairs Aβ clearance and regenerates ROS [62]. Over 20–30 years, this cascade culminates in hippocampal atrophy, synaptic loss, and neuronal death – explaining the delayed clinical onset aligned with epidemiological latency [63].

## Advantages and limitations

This study demonstrates several strengths. First, it pioneers the analysis of GBD 2021 datasets to delineate temporal patterns in DALYs attributable to smoking-related ADD. Through systematic application of advanced analytical approaches, the investigation yields multidimensional insights into sex-specific, demographic, and geographic variations of ADD burden. These evidence-based findings effectively inform targeted intervention strategies across nations at different SDI levels, while establishing a methodological framework for subsequent epidemiological investigations.

However, the study has limitations. First, there is always a problem with the quality, reliability, and scope of data on smoking-related DALYs of ADD. Second, it is difficult to relate our findings to particular smoking initiatives since tobacco control measures are implemented differently in different nations. To learn more about the long-term impacts of smoking on ADD risk and the consequences of tobacco control strategies, future research must concentrate on longitudinal studies. Furthermore, further study is required to comprehend how environmental risk factors and genetic predisposition interact to cause ADD.

## Conclusion

Vigilant intervention was necessary since, despite a decline in the ASDRs of smoking-related ADD burden between 1990 and 2021, the absolute numbers of DALYs continued to rise.

Smoking-related ADD was more common in middle-aged and older adults, men, and those living in high-middle-income nations. Reducing the worldwide burden of smoking-related ADD requires comprehensive tobacco control, public education campaigns, equal access to healthcare, and effective tobacco control policies that are adapted to regional and national circumstances.

## Supporting information

**S1 Table. Age-Specific DALYs rates of AD attributable to Smoking for Female and Male by Global.**
(XLSX)

**S2 Table. Age-Specific DALYs number of AD attributable to Smoking for Female and Male by SDI Region.**
(XLSX)

**S3 Table. Age-Standardised DALYs rates of AD attributable to Smoking for Both sex by 204 Country and territories, 2021.**
(XLSX)

**S4 Table. Age-Standardised DALYs rates and SDI of AD attributable to Smoking for Both sex by 204 Global and territories, from 1990 to 2021.**
(XLS)

**S5 Table. Joinpoint analysis of crude DALYs rate for AD and other dementias attributable to smoking for both sex, in global, grouped by age, from 1990 to 2021.**
(XLS)

**S6 Table. Joinpoint analysis of crude DALYs rate for AD and other dementias attributable to smoking for Female, in global, grouped by age, from 1990 to 2021.**
(XLS)

**S7 Table. Joinpoint analysis of crude DALYs rate for AD and other dementias attributable to smoking for Male, in global, grouped by age, from 1990 to 2021.**
(XLS)

**S8 Table. Joinpoint analysis of ASDR for AD and other dementias attributable to smoking for both sex, grouped by SDI regions, from 1990 to 2021.**
(XLS)

**S9 Table. Joinpoint analysis of ASDR for AD and other dementias attributable to smoking for Female, grouped by SDI regions, from 1990 to 2021.**
(XLS)

**S10 Table. Joinpoint analysis of ASDR for AD and other dementias attributable to smoking for Male, grouped by SDI regions, from 1990 to 2021.**
(XLS)

**S11 Table. Frontier analysis of the ASDR of AD and other dementias attributes to smoking for both sexes by 204 countries and territories, 2021.**
(XLS)

**S12 Table. Age-Standardised DALYs rates and SDI of AD attributable to Smoking for both sexes by 21 regions, from 1990 to 2021.**
(XLS)

## Author contributions

**Conceptualization:** Qifeng Tong.

**Data curation:** Qifeng Tong, Jianing Wu.

**Formal analysis:** Qifeng Tong, Jianing Wu, Hao Wu.

**Funding acquisition:** Qifeng Tong.

**Investigation:** Qifeng Tong, Jianing Wu, Qingchuan Jiao.

**Methodology:** Qifeng Tong, Jianing Wu, Qingchuan Jiao.

**Project administration:** Qingchuan Jiao.

**Resources:** Jianing Wu, Qingchuan Jiao, Hao Wu.

**Software:** Qifeng Tong, Jianing Wu, Hao Wu.

**Supervision:** Jianqiu Gong, Hao Wu, Qiang Wu.

**Validation:** Jianqiu Gong.

**Visualization:** Jianqiu Gong.

**Writing – original draft:** Qifeng Tong, Qiang Wu.

**Writing – review & editing:** Qifeng Tong, Jianqiu Gong, Qiang Wu.

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
