## [Decision Letter · Decision Letter 0]

21 Jul 2025

PONE-D-25-28584Global burden of Alzheimer's disease and other dementias attributable to smoking in 204 countries and territories, 1990–2021PLOS ONE

Dear Dr. Wu,

Thank you for submitting your manuscript to PLOS ONE. After careful consideration, we feel that it has merit but does not fully meet PLOS ONE’s publication criteria as it currently stands. Therefore, we invite you to submit a revised version of the manuscript that addresses the points raised during the review process.

**ACADEMIC EDITOR: While the topic is of interest, the manuscript needs lots of improvement, particularly in the methodology and discussion. My additional comments are also included. **

We look forward to receiving your revised manuscript.

Kind regards,

Thien Tan Tri Tai Truyen, M.D.

Academic Editor

PLOS ONE

Journal Requirements: 

Additional Editor Comments:

1. The abstract needs to be rewritten. Please reduce the conclusion and report more findings in the results which is quite immature at this moment. 

2. Introduction: Lines 69-87: Please reduce this section. You should focus on the current evidence of association between smoking and AD. A further discussion about current national policies, challenges might be useful. Please reduce the discussion about other risk factors which are not the primary objectives of your study. 

3 Methods:

3.1 Data source: A more detailed description about GBD database and IHME including their objectives and primary references is beneficial. 

3.2 Data collection: Please specify your data collection protocol. What are the collected variables and how did you collect them? The link to GBD database is also needed for others scientist replicate or verify your study. 

4. Discussion:

4.1 SDI correlation analysis: AD attributed to smoking is a challenging issue in the majority of developing countries, I suggest you to include recent studies using similar database investigating this condition in developing countries to enhance your context. (suggested references: https://doi.org/10.1016/j.cccb.2025.100390, DOI: 10.1097/MS9.0000000000002344)

Reviewers' comments:

Reviewer's Responses to Questions

**Comments to the Author**

1. Is the manuscript technically sound, and do the data support the conclusions?

Reviewer #1: Yes

Reviewer #2: Yes

Reviewer #3: Yes

2. Has the statistical analysis been performed appropriately and rigorously? 

Reviewer #1: Yes

Reviewer #2: Yes

Reviewer #3: Yes

3. Have the authors made all data underlying the findings in their manuscript fully available?

Reviewer #1: Yes

Reviewer #2: Yes

Reviewer #3: Yes

4. Is the manuscript presented in an intelligible fashion and written in standard English?

Reviewer #1: Yes

Reviewer #2: Yes

Reviewer #3: Yes

5. Review Comments to the Author

Reviewer #1: 1) In the abstract, results paragraph: "From 1990 to 2019…", but in the methods section, the period considered is 1990 to 2021. To be corrected.

2) Line 105: "GBD 2021 to estimate DALY has been thoroughly explained elsewhere." Please provide a reference for that.

3) Lines 119-121: Why use the DSMMD since it is the GBD database that is the reference for indicating whether there is a mental disorder and also indicates whether the impairment is due to smoking. Otherwise, the authors should also explain how ADD is related to tobacco use.

4) Lines 221-223, the sentence is unclear and needs to be reworded: what does "declined in male, female, and both sexes" mean? "Then began to rise" does not indicate when the change from decrease to increase occurs.

5) Line 260 “Meddle-High”, please correct.

6) Line 278-279 “middle SDI countries exposed increasing burdens (EAPC= -0.17%)”. The EAPC=-0.17% is negative and thus is not an increase. Please correct.

7) In the Discussion section Lines 277-278: “high-SDI regions achieved substantial burden reduction (EAPC = -1.34%)”. This needs more explanation with regard to the conclusions of Lines 230-235 (“Across all SDI regions from 1990 to 2021, the relationship showed a bimodal fluctuation upward trend, the ASDRs for ADD initially increased with rising SDI, peaking around an SDI of 0.5, before starting to decline. Subsequently, starting from SDI greater than 0.6, the ASDR value rises again, reaching a second peak at about 0.8, and finally decreasing"). See also Lines 254-255 “countries with higher SDI have greater inconsistency in ASDRs”

Reviewer #2: This manuscript presents a well-executed and globally relevant analysis utilizing GBD 2021 data. The topic is significant, timely, and aligns with the scope of the journal. The study addresses a modifiable risk factor for dementia—smoking—across sociodemographic, temporal, and geographic gradients, adding value to the literature on dementia prevention.However, several minor revisions are necessary to improve the clarity, precision, and presentation of the manuscript. My specific comments are as follows:

1.Introduction:

Briefly discuss limitations of prior studies that focused only on mortality without DALYs, to better justify the contribution.Consider providing a succinct conceptual model or diagram of the mechanisms linking smoking to dementia.

2.Methods:

Definitions of SDI and components could be streamlined or moved to supplementary material.

Clarify the temporal cut-off point: some trend analyses seem to stop at 2019, not 2021—please ensure consistency.

3.Results:

Improve integration of tables/figures into the narrative by referencing specific values and patterns more explicitly.

For Figure 5, further clarify what constitutes “frontier performance” and how "effectiveness differences" were derived.

4.While the discussion comprehensively interprets regional and gender-based disparities in ADD burden, several areas warrant refinement to enhance clarity and scholarly depth:

-Overreliance on Speculation:

The manuscript attributes differences in ASDR trends across countries to policy gaps or underinvestment in health without consistent empirical backing. These claims, while plausible, should be either substantiated by relevant health expenditure or tobacco control implementation data, or more cautiously phrased to reflect inference rather than direct causation.

-Expand Interpretation of Bimodal SDI-ASDR Relationship:

The bimodal association between SDI and dementia burden is intriguing but underexplored. The authors could strengthen this discussion by elaborating on specific health system inefficiencies or diagnostic infrastructure disparities that may explain the secondary peak in high-SDI countries.

-Integrate More Mechanistic Insights:

Although several plausible pathophysiological mechanisms are mentioned (oxidative stress, tau phosphorylation, etc.), this section reads like a literature list. Consider synthesizing these mechanisms into a coherent path model or integrating them with epidemiological implications (e.g., latency effects of smoking on neurodegeneration).

-Policy Implication Section Needs Nuance:

Statements like “even high-SDI countries risk accumulating preventable burden” are important, but could be more persuasive if connected to current challenges in dementia prevention—e.g., limitations of early diagnosis or gaps in long-term care planning.

-Gender Disparities – Expand Interpretative Depth and Policy Implications

The manuscript identifies that male individuals bear a higher crude DALY rate attributable to smoking-related ADD, and that the rate of decline in ASDR is more rapid in females. While the discussion acknowledges biological and behavioral explanations (e.g., smoking prevalence, secondhand smoke, physical activity, dietary patterns), it remains descriptive.

I believe the manuscript can be accepted after addressing the relatively minor but important concerns listed above.

Reviewer #3: Thank you for the invitation to review the manuscript.

The authors used the Global Burden of Disease (GBD), a large-scale disease burden database, to conduct this analysis, exploring the global prevalence of Alzheimer's disease and other types of dementia caused by smoking. This is a very interesting study; however, I have the following suggestions for the authors before this manuscript is formally published.

1. In the introduction, the mechanism by which smoking causes ADD needs to be more clearly defined.

2. In the introduction section, the necessity of conducting this study needs to be supplemented.

3. In the discussion section, please provide more solutions by integrating the mechanisms through which smoking causes ADD.

6. PLOS authors have the option to publish the peer review history of their article (what does this mean? ). If published, this will include your full peer review and any attached files.

**Do you want your identity to be public for this peer review?** For information about this choice, including consent withdrawal, please see our Privacy Policy .

Reviewer #1: **Yes: ** Abdel-Kader Boulanouar

Reviewer #2: **Yes: ** QUYNH PHUONG VO

Reviewer #3: **Yes: ** Xuanjie Chen

---

## [Author Response · Author response to Decision Letter 1]

14 Aug 2025

Additional Editor Comments:

1. The abstract needs to be rewritten. Please reduce the conclusion and report more findings in the results which is quite immature at this moment.

Response: We thank the reviewer for the valuable suggestion. The abstract has been comprehensively rewritten to significantly reduce speculative conclusions and to instead emphasize key findings, including peak ASDR heterogeneity at SDI 0.75 and substantial effectiveness gaps in middle-high SDI nations. Please see Lines 35-51 of the revised manuscript with track changes for details.

2. Introduction: Lines 69-87: Please reduce this section. You should focus on the current evidence of association between smoking and AD. A further discussion about current national policies, challenges might be useful. Please reduce the discussion about other risk factors which are not the primary objectives of your study.

Response: Thank you for this constructive suggestion. We have revised the Introduction (Lines 84-89 and 96-105) as follows:

1. Removed detailed discussions of non-core risk factors (specifically hyperglycemia and high BMI) unrelated to the smoking-AD association.

2. Streamlined focus on current epidemiological evidence linking tobacco use to Alzheimer’s disease.

3. Added a concise analysis of national tobacco control policy gaps (e.g., taxation inadequacies, smoke-free zone enforcement challenges) to strengthen public health context.

These edits refine the section’s alignment with the study’s primary objectives while reducing redundancy.

3. Methods:

3.1. Data source: A more detailed description about GBD database and IHME including their objectives and primary references is beneficial.

Response: We appreciate this constructive suggestion. In the revised Data Source and Collection section (Lines 165-177), we have added the details.

3.2. Data collection: Please specify your data collection protocol. What are the collected variables and how did you collect them? The link to GBD database is also needed for others scientist replicate or verify your study.

Response: We thank you for these valuable remarks. Modifications have been implemented as follows. We have now explicitly defined the protocol. Additionally, we fully agree with the need for data accessibility and have embedded the direct access link in both the Methods and Data Availability sections. Please refer to Lines 177 - 187.

4. Discussion: SDI correlation analysis: AD attributed to smoking is a challenging issue in the majority of developing countries, I suggest you to include recent studies using similar database investigating this condition in developing countries to enhance your context. (suggested references: https://doi.org/10.1016/j.cccb.2025.100390, DOI: 10.1097/MS9.0000000000002344)

Response: Thank you for this valuable suggestion. We agree that investigating smoking-attributable ADD in developing countries is a critical context for our study. To strengthen the background and contextualization of our findings, we have incorporated the recent key studies focusing on this issue in developing countries, as you suggested. Please refer to Lines 527 - 528, Discussion.

Reviewer 1

1. In the abstract, results paragraph: "From 1990 to 2019…", but in the methods section, the period considered is 1990 to 2021. To be corrected.

Response: We sincerely apologize for the oversight regarding the study period in the abstract and results section. The correct study period is indeed 1990 to 2021, as stated in the Methods section. We have updated this period consistently throughout the entire manuscript, including the Abstract, Results, and any other relevant sections.

Furthermore, during our thorough re-examination of the data prompted by your review, we discovered that some data in Table 1 contained errors. We have comprehensively and thoroughly re-checked the raw data used for analysis and confirmed that the underlying source data is accurate and shows no anomalies. We attribute these errors in Table 1 to mistakes that occurred during the process of copying data from the source files into the table. We have now corrected these erroneous entries in Table 1. Corrected data in Table 1 are highlighted in yellow (Revised Manuscript with Track Changes). We have also corrected related misinterpretations in Lines 285-290 of the Results section that stemmed from these initial data errors.

Please accept our apologies for this inadvertent error. We take full responsibility for ensuring the accuracy of all data in this work.

2. Line 105: "GBD 2021 to estimate DALY has been thoroughly explained elsewhere." Please provide a reference for that.

Response: Thank you for pointing this out. We have now added the requested reference (Line 169). As detailed in the cited reference and described in our manuscript, DALY values are calculated for specific combinations of gender, age group, disease, region, and income level using the equation: DALY = Years of Life Lost (YLL) + Years Lived with Disability (YLD). Essentially, DALYs represent the sum of YLLs and YLDs. The GBD study incorporates uncertainty intervals to reflect the precision of its estimates. Each GBD estimate is generated through 1,000 computational iterations. In each iteration, distributions (rather than single point estimates) are sampled for data inputs, data transformations, and model choices. The 95% uncertainty interval is then defined by the 25th and 97.5th ordered values from these 1,000 iterations. Wider uncertainty intervals typically arise from limited data availability, small study sizes, or conflicting data sources. Conversely, narrower intervals result from extensive data, large studies, and consistent data across sources. DALYs are reported per 100,000 population. One DALY equates to the loss of one year of full health. This metric enables the estimation of total healthy life years lost due to specific causes and risk factors at global, regional, national, and sub-national levels (by age, sex, and income).

3. Lines 119-121: Why use the DSMMD since it is the GBD database that is the reference for indicating whether there is a mental disorder and also indicates whether the impairment is due to smoking. Otherwise, the authors should also explain how ADD is related to tobacco use.

Response: We appreciate the reviewer's insightful comment. The use of both Diagnostic and Statistical Manual of Mental Disorders (DSM) criteria and GBD data is intentional and complementary. In fact, the abbreviated definition for AD on the GBD website are precisely based on the DSM. (https://www.healthdata.org/research-analysis/diseases-injuries-risks/factsheets/2021-alzheimers-disease-and-other-dementias)

4. Lines 221-223, the sentence is unclear and needs to be reworded: what does "declined in male, female, and both sexes" mean? "Then began to rise" does not indicate when the change from decrease to increase occurs.

Response: We sincerely appreciate the reviewer’s insightful feedback regarding clarity in this sentence. We agree that the original phrasing was ambiguous in two key aspects:

1. The listing of "male, female, and both sexes" during the decline phase was redundant, as a decline in both individual sexes inherently implies a decline in the combined group. This change trend has already been described in Lines 406-409, and therefore we have removed it.

2. "Then began to rise" indicates that 2019 is when the change from decrease to increase occurs. Please refer to Lines 408-409.

5. Line 260 “Meddle-High”, please correct.

Response: Thank you for pointing out this typographical error. The term "Meddle-High" on line 260 has been corrected to "Middle-High" in the revised manuscript. Please refer to Lines 463, Result.

6. Line 278-279 “middle SDI countries exposed increasing burdens (EAPC = -0.17%)”. The EAPC = -0.17% is negative and thus is not an increase. Please correct.

Response: Thank you for catching this important discrepancy. You are absolutely correct that a negative EAPC (-0.17%) indicates a decrease, not an increase. The sentence has been revised to accurately reflect the trend: "middle SDI countries exhibited a modest decline in burdens (EAPC = -0.17%)". We apologize for the error and have carefully verified the data and interpretation throughout the manuscript. Please refer to Lines 484, Discussion.

7. In the Discussion section Lines 277-278: “high-SDI regions achieved substantial burden reduction (EAPC = -1.34%)”. This needs more explanation with regard to the conclusions of Lines 230-235 (“Across all SDI regions from 1990 to 2021, the relationship showed a bimodal fluctuation upward trend, the ASDRs for ADD initially increased with rising SDI, peaking around an SDI of 0.5, before starting to decline. Subsequently, starting from SDI greater than 0.6, the ASDR value rises again, reaching a second peak at about 0.8, and finally decreasing"). See also Lines 254-255 “countries with higher SDI have greater inconsistency in ASDRs”

Response: We sincerely appreciate your valuable feedback. We apologize for any confusion caused by our initial description and have revised address this.

The apparent contradiction can be explained by intra-group heterogeneity within high-SDI regions, as shown in Figure 5A. While the overall trend shows a secondary ASDR peak around SDI 0.8, this masks significant regional variations:

1. Divergent trajectories: Western Europe, high-income Asia Pacific, and North American regions exhibited rapid ASDR declines, while Eastern Europe (with comparable SDI) showed minimal decline

2. Methodological consideration: Our SDI-ASDR analysis used data from 21 regions rather than aggregated SDI quintiles, which may contribute to observed heterogeneities

We have incorporated a detailed explanation of heterogeneity in the manuscript. Please refer to Lines 432-435 and Lines 454-458.

Reviewer 2:

1. Introduction: Briefly discuss limitations of prior studies that focused only on mortality without DALYs, to better justify the contribution. Consider providing a succinct conceptual model or diagram of the mechanisms linking smoking to dementia.

Response: Thank you for highlighting this crucial aspect. Prior studies focusing solely on mortality metrics significantly underestimate the true burden of smoking-attributable dementia. Mortality data fail to capture the substantial non-fatal health loss inherent to dementia, which manifests as prolonged disability, profound cognitive decline, and immense caregiver burden. By omitting DALYs, these studies provide an incomplete picture, overlooking the disease's major socioeconomic impact and hindering accurate assessment of prevention needs. To address this limitation and comprehensively quantify the burden, our study employs DALYs, integrating both fatal and non-fatal outcomes. Please refer to Lines 105-107, Introduction.

2. Methods: Definitions of SDI and components could be streamlined or moved to supplementary material. Clarify the temporal cut-off point: some trend analyses seem to stop at 2019, not 2021—please ensure consistency.

Response: We thank the reviewer for the constructive suggestion to streamline the definition of the SDI. We have now clearly defined the SDI components in a more concise manner while retaining all essential methodological information. Please refer to Lines 195-197. Additionally, we apologize for the oversight. The correct study period is 1990 to 2021 as stated in Methods. This has been updated throughout the manuscript, including the Abstract and Results. Thank you for your careful review.

3. Results: Improve integration of tables/figures into the narrative by referencing specific values and patterns more explicitly. For Figure 5, further clarify what constitutes “frontier performance” and how "effectiveness differences" were derived.

Response: We sincerely appreciate these constructive suggestions. We have referenced specific values in the results section to make it more explicitly.

We thank the reviewer for the request to clarify the concepts of "frontier performance" and "effectiveness differences". We have revised the Methods section (Lines 260-264) to provide explicit definitions and computational details as follows:

1. Definition of Frontier Performance: Frontier performance represents the theoretical minimum burden of disease achievable for a given SDI level. Using frontier analysis (specifically, stochastic frontier analysis), we identified the optimal performance frontier by fitting a regression model to the lowest observed ASDR across different SDI levels. This frontier serves as a benchmark for the best achievable health outcomes given a region's development status.

2. Calculation of Effectiveness Differences: Effectiveness differences quantify the gap between observed performance and the frontier benchmark. For each location, it was computed as: effectiveness differences are equal to the value of the ASDR for each country minus the value of the frontier, adjusted for SDI.

4. While the discussion comprehensively interprets regional and gender-based disparities in ADD burden, several areas warrant refinement to enhance clarity and scholarly depth:

4.1. Overreliance on Speculation: The manuscript attributes differences in ASDR trends across countries to policy gaps or underinvestment in health without consistent empirical backing. These claims, while plausible, should be either substantiated by relevant health expenditure or tobacco control implementation data, or more cautiously phrased to reflect inference rather than direct causation.

Response: We sincerely appreciate this valuable critique regarding causal attribution in our discussion of ASDR disparities. We agree that inferring direct causation from the available information requires caution. We have carefully revised the relevant wording in the discussion, adopting more cautious phrasing. Please refer to Lines 524-562, Discussion.

4.2. Expand Interpretation of Bimodal SDI-ASDR Relationship: The bimodal association between SDI and dementia burden is intriguing but underexplored. The authors could strengthen this discussion by elaborating on specific health system inefficiencies or diagnostic infrastructure disparities that may explain the secondary peak in high-SDI countries.

Response: We have expanded the discussion of diagnostic policy challenges in high-SDI countries (Lines 573-595). Our analysis shows that:

1. Resources aren't always well used: Even with enough specialists (e.g., 13.6 per 100,000 people in Sweden), disconnected healthcare systems often lead to long waits (21-70 months).

2. New tools aren't fully adopted: Guidelines recommend biomarkers for diagnosis, but real-world use faces hurdles like PET scanner shortages (UK) and limited capacity (linked to 30% of delays in Germany).

3. Access differs in practice: Despite universal coverage goals, service availability varies between regions and insurance types (as seen in Germany).

4.3. Integrate More Mechanistic Insights: Although several plausible pathophysiological mechanisms are mentioned (oxidative stress, tau phosphorylation, etc.), this section reads like a literature list. Consider synthesizing these mechanisms into a coherent path model or integrating them with epidemiological implications (e.g., latency effects of smoking on neurodegeneration).

Response: We sincerely appreciate these constructive suggestions. We have clarified the pathophysiology showing how smoking-induced oxidative stress initiates Aβ/tau pathology, amplified by neuroinflammation, culminating in neurodegeneration after 20-30 years – thereby resolving mechanistic and epidemiological observations simultaneously. Please refer to Lines 647 - 667, Discussion.

4.4. Policy Implication Section Needs Nuance: Statements like “even high-SDI countries risk accumulating preventable burden” are important but could be more persuasive if connected to current challenges in dementia prevention—e.g., limitations of early diagnosis or gaps in long-term care planning.

Response: We thank the reviewer for this valuable suggestion. As requested, we have strengthened the link between preventable dementia burden and current prevention challenges in high-SDI settings. Specifically, we revised the discussion on Line 377-382 to explicitly state that diag

---

## [Decision Letter · Decision Letter 1]

25 Sep 2025

PONE-D-25-28584R1Global burden of Alzheimer's disease and other dementias attributable to smoking in 204 countries and territories, 1990–2021PLOS ONE

Dear Dr. Wu,

Thank you for submitting your manuscript to PLOS ONE. After careful consideration, we feel that it has merit but does not fully meet PLOS ONE’s publication criteria as it currently stands. Therefore, we invite you to submit a revised version of the manuscript that addresses the points raised during the review process.

**ACADEMIC EDITOR: The manuscript has improved significantly. However, please carefully review it again for grammatical errors and duplicated content (e.g., lines 309–311 are repeated in lines 312–314). **==============================

We look forward to receiving your revised manuscript.

Kind regards,

Thien Tan Tri Tai Truyen, M.D.

Academic Editor

PLOS ONE

Journal Requirements:

Reviewers' comments:

Reviewer's Responses to Questions

**Comments to the Author**

1. If the authors have adequately addressed your comments raised in a previous round of review and you feel that this manuscript is now acceptable for publication, you may indicate that here to bypass the “Comments to the Author” section, enter your conflict of interest statement in the “Confidential to Editor” section, and submit your "Accept" recommendation.

Reviewer #1: (No Response)

Reviewer #2: All comments have been addressed

Reviewer #3: All comments have been addressed

2. Is the manuscript technically sound, and do the data support the conclusions?

Reviewer #1: Partly

Reviewer #2: Yes

Reviewer #3: Yes

3. Has the statistical analysis been performed appropriately and rigorously? 

Reviewer #1: No

Reviewer #2: Yes

Reviewer #3: Yes

4. Have the authors made all data underlying the findings in their manuscript fully available?

Reviewer #1: (No Response)

Reviewer #2: Yes

Reviewer #3: (No Response)

5. Is the manuscript presented in an intelligible fashion and written in standard English?

Reviewer #1: Yes

Reviewer #2: Yes

Reviewer #3: Yes

6. Review Comments to the Author

Reviewer #1: 1) The changes to the manuscript announced in the response letter cannot be found in the corrected manuscript using the line numbers indicated as containing these changes. Examining the authors' responses is therefore difficult, if not impossible.

2) Upon reexamining the data, the authors discovered a large number of errors (over sixty) in Table 1. The authors do not state what led them to reexamine the data. They also state, on the one hand, that "We have comprehensively and thoroughly rechecked the raw data used for analysis and confirmed that the underlying source data is accurate and shows no anomalies," and on the other hand, they had to correct "related misinterpretations in Lines 285-290 of the Results section that stemmed from these initial data errors." It is therefore unclear whether the errors had an impact on the interpretation of the results.

3) The main contribution of this manuscript is the consideration of DALY. This term therefore deserves to be better defined than by a simple equality which is not very understandable. The determination of DALY is more complex. Examples of calculation on concrete cases showing how the 2 terms (YLD and YLL) are obtained would be more informative. The interest of these examples would be to show how (and to what extent) the consideration of DALYs modifies the data.

Reviewer #2: I have reviewed the revised manuscript and confirm that the authors have adequately addressed my previous comments. Key improvements include clearer justification in the Introduction regarding the use of DALYs, concise SDI definitions, improved integration of tables and figures, and clearer explanation of “frontier performance” and “effectiveness differences.” The Discussion has been notably strengthened, with more cautious causal language, expanded interpretation of the SDI-ASDR relationship, and deeper exploration of mechanistic and gender-based differences. The manuscript has improved in clarity, methodological rigor, and policy relevance. No further concerns.

Reviewer #3: All authors have refined all the suggestions according to the reviewers' comments and have responded appropriately.

7. PLOS authors have the option to publish the peer review history of their article (what does this mean? ). If published, this will include your full peer review and any attached files.

**Do you want your identity to be public for this peer review?** For information about this choice, including consent withdrawal, please see our Privacy Policy .

Reviewer #1: No

Reviewer #2: **Yes: ** VO PHUONG QUYNH

Reviewer #3: No

---

## [Author Response · Author response to Decision Letter 2]

29 Sep 2025

Response letter to Reviewers by Dr. Wu et al titled “Global burden of Alzheimer's disease and other dementias attributable to smoking in 204 countries and territories, 1990–2021”

PONE-D-25-28584

Dear. Editor and Reviewers,

Thank you very much for your interest in our manuscript entitled “Global burden of Alzheimer's disease and other dementias attributable to smoking in 204 countries and territories, 1990–2021”. To address the comments and concerns the Reviewers have raised, we have included a point-by-point Response to each comment. Within the revised manuscript, changes to the text in Response to the comments are marked as red font. And we revised the manuscript style in accordance with the PLOS ONE.

We appreciate the suggestions and comments by the Reviewers. As a consequence of these valuable suggestions, we believe that our manuscript has been much strengthened.

Please feel free to contact me if you and reviewers have further suggestions. I would be happy to make a revision according to their comments.

Yours sincerely

Dr. Wu

E-mail: dr5rehab@163.com

Additional Editor Comments:

1) The manuscript has improved significantly. However, please carefully review it again for grammatical errors and duplicated content (e.g., lines 309–311 are repeated in lines 312–314)

Response:

Thank you for your thoughtful feedback. I have carefully reviewed the entire manuscript and made thorough revisions to address the grammatical errors throughout the text. Additionally, I have removed the duplicated content, specifically the repeated sentences (lines 337–339 in Revised Manuscript with Track Changes), to improve the clarity and flow of the manuscript. Your comments have been immensely helpful in enhancing the quality of this work.

Reviewer 1

1) The changes to the manuscript announced in the response letter cannot be found in the corrected manuscript using the line numbers indicated as containing these changes. Examining the authors' responses is therefore difficult, if not impossible.

Response:

I sincerely apologize for the incorrect line numbers I provided in the previous response; I realize this oversight made your second review far more difficult than it should have been. Although the revised manuscript has already been evaluated, I take full responsibility for the mistake and will implement a rigorous cross-checking procedure for all future submissions to ensure that every change is referenced accurately.

Thank you very much for your patience and for helping us improve the work.

2) Upon reexamining the data, the authors discovered a large number of errors (over sixty) in Table 1. The authors do not state what led them to reexamine the data. They also state, on the one hand, that "We have comprehensively and thoroughly rechecked the raw data used for analysis and confirmed that the underlying source data is accurate and shows no anomalies," and on the other hand, they had to correct "related misinterpretations in Lines 285-290 of the Results section that stemmed from these initial data errors." It is therefore unclear whether the errors had an impact on the interpretation of the results.

Response:

We sincerely apologize for the confusion caused by the inconsistencies. After the previous round of review drew our attention to a handful of minor descriptive figures elsewhere in the manuscript, we voluntarily undertook a line-by-line re-check of every numeric entry to ensure no similar transcription errors had occurred in any other part of the paper. During this self-initiated audit—conducted independently by three co-authors—we discovered that some values in Table 1 had been transposed (e.g., DALYs for two regions were swapped) when the analyzed data were copied from R into the manuscript. These mistakes were purely transcriptional; the underlying source files extracted from the GBD database remain intact and unchanged.

We immediately corrected Table 1 and the corresponding sentences (lines 218–226 in this version of Revised Manuscript with Track Changes) in the revised manuscript submitted last round. We take full responsibility for this mistake and are deeply sorry. Fortunately, all subsequent analyses were conducted directly on the raw source files, so all figures and remaining reported results are correct.

To remove any remaining doubt, we now provide the raw datasets in supplementary materials:

Smoking_AD.csv: The original data downloaded from the GBD database includes locations such as the world, five SDI regions, seven super-regions, and 21 regions. The years range from 1990 to 2021.

Smoking_AD_country.csv: The "location" field covers 204 countries and regions.

Smoking_AD_21Regions.csv: The ASDRs values for 21 regions in 1990 and 2021.

Smoking_AD_SuperRegionandGlobal.csv: The ASDRs values for the global level, 5 SDI regions, and 7 super regions in 1990 and 2021.

Thank you for your patience and for holding us to the highest standards; we have instituted a mandatory double-entry verification protocol for all numeric content in future submissions.

3) The main contribution of this manuscript is the consideration of DALY. This term therefore deserves to be better defined than by a simple equality which is not very understandable. The determination of DALY is more complex. Examples of calculation on concrete cases showing how the 2 terms (YLD and YLL) are obtained would be more informative. The interest of these examples would be to show how (and to what extent) the consideration of DALYs modifies the data.

Response:

Thank you very much for highlighting the need to clarify DALY. We have now added a concise, step-by-step example (50-year-old smoker who develops AD and dies at 65) that shows how YLL and YLD are obtained and how their sum changes the perspective compared with mortality alone. We stress that real-world figures are produced with the full GBD modelling machinery and hope this revision makes the concept accessible to readers. We appreciate your guidance in improving the manuscript.

Please refer to Lines 128-144 in Revised Manuscript with Track Changes.

---

## [Editor Report · Decision Letter 2]

1 Oct 2025

Global burden of Alzheimer's disease and other dementias attributable to smoking in 204 countries and territories, 1990–2021

PONE-D-25-28584R2

Dear Dr. Wu,

We’re pleased to inform you that your manuscript has been judged scientifically suitable for publication and will be formally accepted for publication once it meets all outstanding technical requirements.

Kind regards,

Thien Tan Tri Tai Truyen, M.D.

Academic Editor

PLOS ONE
---

## [Editor Report · Acceptance letter]

PONE-D-25-28584R2

PLOS ONE

Dear Dr. Wu,

I'm pleased to inform you that your manuscript has been deemed suitable for publication in PLOS ONE. Congratulations! Your manuscript is now being handed over to our production team.

Kind regards,

on behalf of

Dr. Thien Tan Tri Tai Truyen

Academic Editor

PLOS ONE